# A space-based quantum gas laboratory at picokelvin energy scales

Naceur Gaaloul [1,12] ✉, Matthias Meister [2,12], Robin Corgier[1,3,11], Annie Pichery[1,3], Patrick Boegel [4], Waldemar Herr [5], Holger Ahlers[1,5], Eric Charron [3], Jason R. Williams[6], Robert J. Thompson [6], Wolfgang P. Schleich [2,4,7,8,9], Ernst M. Rasel[1] & Nicholas P. Bigelow [10] ✉

Ultracold quantum gases are ideal sources for high-precision space-borne sensing as proposed for Earth observation, relativistic geodesy and tests of fundamental physical laws as well as for studying new phenomena in many-body physics during extended free fall. Here we report on experiments with the Cold Atom Lab aboard the International Space Station, where we have achieved exquisite control over the quantum state of single $^{87}$Rb Bose-Einstein condensates paving the way for future high-precision measurements. In particular, we have applied fast transport protocols to shuttle the atomic cloud over a millimeter distance with sub-micrometer accuracy and subsequently drastically reduced the total expansion energy to below 100 pK with matter-wave lensing techniques.

Very low energy scales in physical systems enable quantum physicists to explore new states of matter, observe phase transitions and accurately measure fundamental physical quantities. Nowadays, experiments with ultracold quantum gases[1,2] allow researchers to routinely access temperature scales well below a nanokelvin. However, the ultimate potential of these atomic systems, for example in quantum sensing[3], only manifests itself when they are probed in free evolution and for long observation times of several seconds. Carrying out experiments in space, where quantum matter can freely float for unrivaled extended times, is therefore an active research quest in physics, which has seen recent advancement by the pioneering generation of atomic Bose-Einstein condensates (BECs) in space[4] and in orbit[5]. In addition to extended free fall times, the space environment allows the straightforward manipulation of quantum gases or mixtures

of them by undistorted symmetric traps enabling for example the creation of shell topologies[6] or uniform Bose gases[7].

We have performed a series of experiments employing NASA's Cold Atom Lab (CAL)[5] aboard the International Space Station (ISS), where we have created and manipulated BECs that expand with energies of a few tens of picokelvin. Our methods promote the CAL multi-user facility into a high-precision, in-orbit laboratory accelerating numerous research applications such as the study of quantum bubbles[6,8–10], space atom lasers[11–13], few-body physics[14], quantum reflection[15] from material surfaces or entangled state preparation[16,17]. Moreover, this exquisite control will be crucial for space-based quantum information processing[18], quantum simulation[19], quantum communication[20] and will boost applications such as Earth observation, relativistic geodesy and tests of fundamental physical laws[21–23].

[1]Leibniz University Hannover, Institute of Quantum Optics, QUEST-Leibniz Research School, Hanover, Germany. [2]German Aerospace Center (DLR), Institute of Quantum Technologies, Ulm, Germany. [3]Université Paris-Saclay, CNRS, Institut des Sciences Moléculaires d'Orsay, F-91405 Orsay, France. [4]Institut für Quantenphysik and Center for Integrated Quantum Science and Technology (IQST), Ulm University, Ulm, Germany. [5]Deutsches Zentrum für Luft- und Raumfahrt e. V. (DLR), Institut für Satellitengeodäsie und Inertialsensorik (SI), Callinstraße 30b, 30167 Hannover, Germany. [6]Jet Propulsion Laboratory, California Institute of Technology, Pasadena, CA, USA. [7]Hagler Institute for Advanced Study, Texas A&M University, College Station, TX, USA. [8]Texas A&M AgriLife Research, Texas A&M University, College Station, TX, USA. [9]Institute for Quantum Science and Engineering (IQSE), Department of Physics and Astronomy, Texas A&M University, College Station, TX, USA. [10]Department of Physics and Astronomy, University of Rochester, Rochester, NY 14627, USA. [11]Present address: LNE-SYRTE, Observatoire de Paris, Université PSL, CNRS, Sorbonne Université 61 avenue de l'Observatoire, 75014 Paris, France. [12]These authors contributed equally: Naceur Gaaloul, Matthias Meister. ✉e-mail: gaaloul@iqo.uni-hannover.de; nicholas.bigelow@rochester.edu

Technological progress in the field of cold atoms made it possible to overcome restrictions posed by space platforms such as payload compactness, the need for autonomous operation, the changing orientation and orbital environment during the flight, and to successfully launch cold atom experiments into space. However, to take full advantage of a space-based quantum gas source such as CAL and to support the entire range of possible research applications, exquisite control of the quantum state is required. A first aspect of this control is the ability to transport the created ensemble in a fast and robust fashion, yet realizing an accurate final positioning of the transported atoms[24]. Equally important is to ensure a release from the final trap with a well-controlled and tunable center-of-mass velocity to realize free-space or guided interferometry or generate complex many-body quantum systems such as squeezed or molecular states. Controlling the kinematics and expansion of a quantum gas is crucial towards precision measurements in space. In atom interferometric experiments in particular, the uncertainty about the initial position and initial velocity couples to other quantities and leads to systematic effects such as gravity-gradient-induced ones or Coriolis effects[25]. These systematic shifts are leading sources of uncertainty limiting the performance of state-of-the-art experiments such as tests of the Einstein equivalence principle[26]. Finally, the atomic ensemble's expansion rate needs to be drastically slowed down to kinetic temperatures of less than 100 pK to allow for a detection after seconds-long free drift necessary for pushing the sensitivity of space quantum sensors beyond the performance of their Earth-bound counterparts.

Here we report on important breakthroughs covering the three aspects of quantum state engineering allowing us to accurately control the position, release velocity and expansion rate of a quantum gas ultimately reaching expansion energies of a few tens of picokelvin. Furthermore, we take advantage of the microgravity environment to simplify quantum control protocols allowing the implementation of ab-initio, atom chip-based quantum engineering recipes. We do so using a non-custom multi-user facility enabling the metrological exploitation of a cold quantum gas space-platform through the generation of a robust, stable, and tunable atomic source. These results are aligned with the most challenging requirements of future space-based experiments in the field of fundamental physics[27], geodesy[28] and detection of gravitational waves[29].

## Results

### Experimental setup

The experiments reported here were designed by the Consortium for Ultracold Atoms in Space (CUAS) and executed using CAL, a multi-user quantum gas facility aboard the ISS. The apparatus is described in detail in ref. 5 and features the generation of a $^{87}$Rb BEC by an atom chip-based[30–32] payload. To reduce the complexity of the trapping potential for subsequent experiments, after the condensation the facility-provided quantum gas is transferred from the five-current evaporation trap to a simplified trap generated by the combination of magnetic fields resulting solely from two control currents: $I_{chip}$ is sent through a Z-shaped wire on the atom chip ($x$-$y$-plane) and $I_{coil}$ is injected in an external pair of Helmholtz bias coils oriented along the $y$-axis (parallel to the chip surface). This basic configuration creates a magnetic quadrupole field in the $y$-$z$-plane which is closed in the $x$-direction by the bent tails of the Z-shaped chip wire leading to an approximately harmonic confinement for the atoms. Figure 1a (left) illustrates this setup with the initial BEC made of a few thousand atoms trapped 267 μm away from the atom chip. The BEC oscillates in this trap at angular frequencies $(\omega_x, \omega_y, \omega_z) = 2\pi \cdot$ (29.3, 922, 926) Hz obtained via a complete modeling of all magnetic fields and verified by scanning the hold time $t_{hold,i}$ (see Fig. 1c). The in-trap cloud oscillation amplitude in the $z$-direction amounts to $0.22 \pm 0.05$ μm, which corresponds to a maximum in-trap velocity of $1.3 \pm 0.3$ mm s$^{-1}$.

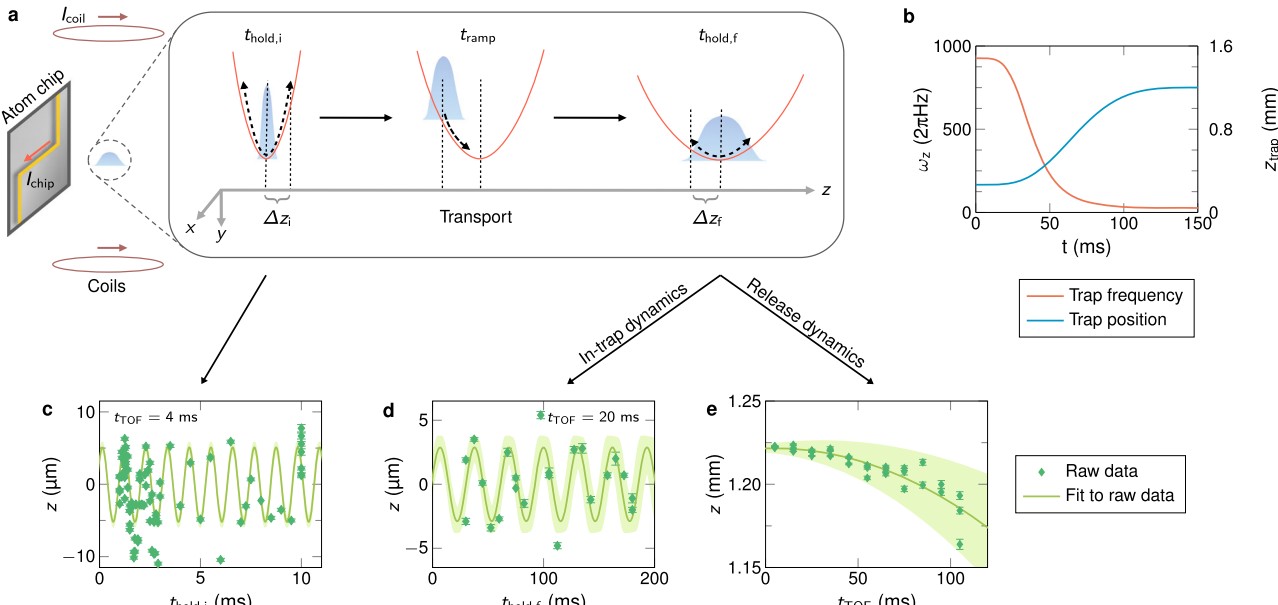

**Fig. 1 | Precise atom chip-aided preparation and characterization of a quantum gas in orbit. a** Fast transport in the $z$-direction away from the atom chip of an initially tightly confined, oscillating $^{87}$Rb BEC to a weakly confining final trap within a total time $t_{ramp}$. **b** Applied ramp of the frequency $\omega_z$ (red) and trap minimum position $z_{trap}$ (blue) during a typical transport. Exemplary center-of-mass motion (green diamonds) of the BEC after time-of-flight $t_{TOF}$ recorded for the initial trap for varying hold times $t_{hold,i}$ (**c**) and after the transport for varying hold times $t_{hold,f}$ (**d**) or by scanning the free expansion time $t_{TOF}$ after release (**e**). The initial sloshing (**c**) is caused by transfer from the facility trap to a simplified two-current ramp trap which is generated by the currents $I_{coil}$ and $I_{chip}$ applied to a pair of Helmholtz coils and a z-shaped chip wire (**a**), respectively. The final sloshing (**d**) is a measure of the quality of our transport with low amplitudes indicating good performance. Residual magnetic field gradients accelerate the atoms in the $F = 2$, $m_F = 2$ quantum state during free expansion (**e**). Fitting (solid lines) a sine (**c**, **d**) or a parabola (**e**) yields the oscillation amplitudes or the release velocity, respectively. Green shaded areas show the 1σ-confidence bounds of the fits and error bars reflect the single-shot detection noise. For more details on the data extraction procedure see the Methods section.

## Bose-Einstein condensate shuttling, positioning and release

Shuttling quantum ensembles is an essential ingredient in operating quantum sensors[3], processing quantum information[18] or conducting secure quantum communication[20,33,34]. Moreover, it is generally beneficial to execute these operations at a position with better optical access and cleaner electromagnetic environment than the preparation area. In the case of an atom chip experiment, it is necessary to move the BEC away from the chip surface, especially given that the released quantum gas can expand up to several millimeters in size, potentially colliding with the chip surface. Moreover, to enable atom optics experiments involving external light beams such as atom interferometry[35], this displacement is required to avoid obstruction of the laser beams by the chip structures compromising the contrast of the interferometer[36]. While in principle a slow, fully adiabatic atom transport could be used, it is crippling due to the accumulation of decoherence effects during transport or simply because it would increase the experimental cycle duration, which could limit a repetitive high-precision metrology experiment. In our system, a sufficiently adiabatic trap minimum shift by 1 mm would take more than 100 s.

Here we realize a fast atomic transport exploiting shortcut-to-adiabaticity methods[37] based on reverse-engineering quantum control protocols as detailed in ref. 38. The method determines a classical trajectory for the atoms, according to fixed boundary conditions, which must be fulfilled experimentally to ensure an optimal transport. These boundary conditions are chosen to guarantee, initially and finally, the center of mass of the cloud being at rest, at the position of the minimum of the trap. This leads to an atomic position following a polynomial function in time. The trap dynamics are found thereafter by inverting Newton's equations (see the Methods sections on shortcut-to-adiabaticity protocols and transport theory).

These protocols greatly benefit from microgravity conditions. On Earth, gravity couples the different degrees of freedom limiting the protocol efficiency[39], imposing different more complicated optimal control solutions[40] and the resulting derivation of complex multi-dimensional control sequences. This, in turn, would require a larger number of experimental control parameters potentially making the implementation impractical. In contrast, in the absence of gravitational acceleration and by employing the simplified two-current trapping scheme our space-based implementation of a shuttling ramp merely relies on the control of one parameter, namely the current of the bias coil $I_{coil}$ which defines the minimum position of the trap with respect to the chip surface. The time dependency of this bias coil current is theoretically determined by an ab-initio approach similar to the one of ref. 38 and a robustness study relying solely on few calibrated values of the trap frequency as a function of the distance to the chip (see the Methods section).

We demonstrate our shuttling approach by considering two different final trap configurations that we denote trap A and trap B. Trap A, featuring a modest transport distance of 0.42 mm triggered by a decrease of the current in the bias coil by a factor 3 during 100 ms, possesses trap frequencies of $(\omega_x, \omega_y, \omega_z) = 2\pi \cdot (25.2, 109, 110)$ Hz making it ideal for matter-wave lensing experiments[38,41] because the almost cylindrical symmetry enables the simultaneous collimation of two spatial dimensions at once. Trap B, reached decreasing the above-mentioned current by a factor of 6 for a total time of 150 ms, corresponds to a long transport distance of 0.93 mm realizing a space-characteristic weak final 3D trapping of $(\omega_x, \omega_y, \omega_z) = 2\pi \cdot (14.4, 35.1, 26.9)$ Hz. This shallow final trap, possible in space because of the weak residual gravitational pull of Earth, is perfect for releasing a quantum gas as it minimizes its expansion energy and eventual parasitic kicks due to technical imperfections of the switch-off. The full details of the time-dependent ramp sequences are displayed in Supplementary Fig. 3.

The short duration of the ramps results in an out-of-equilibrium transport of the quantum gas as illustrated in Fig. 1a (middle). Two

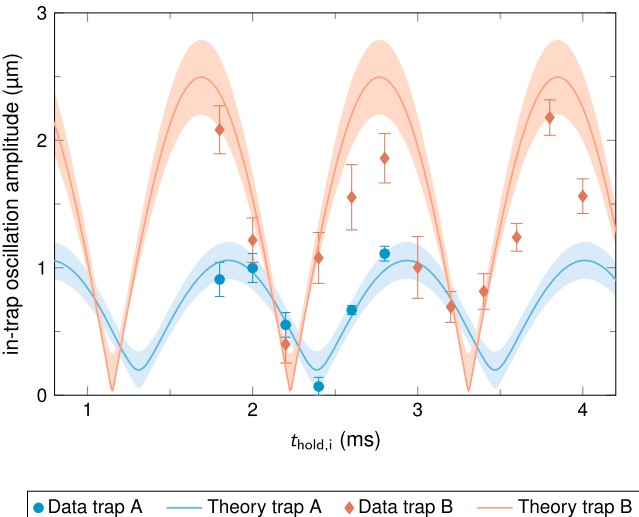

**Fig. 2 | Realization of a sub-μm residual oscillation amplitude of a transported BEC.** Residual in-trap oscillation amplitude of a BEC after transport as a function of the initial hold time $t_{hold,i}$ for two different transport ramps A (blue) and B (red). In sequence A (B) the atoms were transported along the $z$-direction over 0.42 (0.93) mm in 100 (150) ms from an initial trap of frequency $\omega_z = 2\pi \cdot 926$ Hz with initial oscillation amplitude of $0.22 \pm 0.05$ μm to a final trap with frequency $\omega_z = 2\pi \cdot 110$ (26.9) Hz. Transport B is space-enabled due to very weak final 3D trapping. After the transport residual oscillation amplitudes as low as $0.068 \pm 0.072$ μm (trap A) and $0.40 \pm 0.15$ μm (trap B) were measured. Changing the amplitude from 0.2 μm to 0.068 (0.40) μm while lowering the trap frequency by a factor of 8.4 (34) corresponds to a reduction of the oscillation energy by a factor of 738 (359) for trap A (B), respectively. The data (blue circles and red diamonds) is consistent with an ab-initio theoretical model (blue and red lines) based on a chip simulation and mean-field BEC dynamics (see Methods). The error bars and shaded areas show the 1σ-confidence bounds of the fits and the model, respectively.

types of measurements are performed to evaluate the shuttling quality: (i) we record the in-trap oscillations of the transported BEC by varying the hold time $t_{hold,f}$ and imaging the atomic cloud after 20 ms time-of-flight for every data point (Fig. 1d), (ii) we choose one hold time and collect a series of images for a varied time-of-flight (Fig. 1e).

The first type of experiments confirms the successful implementation of the shuttling protocol. As shown in Fig. 2, the residual oscillation amplitudes could be reduced to a minimum of $0.068 \pm 0.072$ μm in the case of ramp A (final trap frequency of $\omega_z = 2\pi \cdot 110$ Hz) and $0.40 \pm 0.15$ μm for ramp B (final trap frequency of $\omega_z = 2\pi \cdot 26.9$ Hz). In order to validate our method, the hold time before the ramp $t_{hold,i}$ is scanned and we obtain a good agreement with our theoretical expectations represented by the solid curves in Fig. 2. The implemented protocol consequently led to a transport of the quantum state over a distance of roughly 1500 times its characteristic size with a residual oscillation amplitude of about 5% of its spatial extent. The transport ramp was implemented using a duration as short as 4 motional cycles of the final trap B period, accomplishing a reduction in the oscillation energy of more than two orders of magnitude compared to the initial facility-provided state.

The positioning uncertainty achieved here will be instrumental in a wide range of sensing experiments such as quantum tests of general relativity that require control of the initial positions of test masses to better than a one micron level[42]. The shuttling performance is also dramatically more advanced than prior transport conducted with neutral atoms on ground[37] making our source suitable for quantum information processing, for example. This latter application, remarkably implemented so far with systems consisting of few ions, requires meticulous positioning uncertainties to keep excitations at the level of a fraction of the motional quantum of the final trap. It is worth noticing

that our transport is at the same level of performance and precision as seminal experiments implemented in quantum information processing with ions[43,44] despite the shuttling starting point in this facility being a BEC of a few thousand atoms in a motionally excited state, as compared to starting with a single-atom quantum system in its ground state, as is true for ion transport experiments. Furthermore, the trap frequencies controlling the shuttling ramps are not constant, but decrease instead by about two orders of magnitude throughout the transport. A final complexity of our transport is the coupling of the three spatial dimensions by the inter-atomic interactions in the $^{87}$Rb BEC.

The second aspect of quantum-state engineering that we perform is about controlling the release velocity of the BEC. Figure 3 shows this velocity recorded again as a function of the initial hold time. Several conclusions can be drawn from this graph: (i) by scanning the initial hold time, it is possible to realize an atomic source with a tunable center-of-mass release velocity which agrees well with the oscillatory behavior of the theoretical prediction. The error bars represent experimental noise and stem from the rather low statistics due to constraints on the total available measurement time (see Supplementary Fig. 4); (ii) a systematic momentum kick is communicated by the chip trap while switching it off, given by almost 1 mm s$^{-1}$ for trap A and few tens of μm s$^{-1}$ in the weak trap configuration B. This momentum kick is typical for atom chips and originates from the slower switch-off behavior of the coil due to its larger inductance compared with the chip structure leading to a slight shift of the trap minimum during release that accelerates the atoms. The magnitude of the kick scales with the original trap frequency in consistency with the measurements; (iii) for a hold time of 2.8 ms the space-enabled ramp B allows the release of BECs with velocities of 35 ± 117 μm s$^{-1}$. By taking the root mean square of the error bars for all hold times of ramp B, we obtain an overall uncertainty of 233 μm s$^{-1}$ which reduces to 165 μm s$^{-1}$ when weighting the individual error bars by the confidence bounds of the theory model (see Supplementary Fig. 4). This uncertainty is close to the one achieved by some of the most advanced metrological

measurements in the field of atom interferometry[26,45] despite the limited number of experiments and the implementation in a multi-user facility. The control of the uncertainty in the initial velocity being comparable to state-of-the-art metrology experiments holds the promise that compact, space-borne laboratories such as CAL are mature for satellite quantum sensing as planned for general relativity tests or Earth observation missions.

## Reaching pK energy scales

Having optimized the kinematics of the atomic source, it is possible to engineer the quantum state of our degenerate gas source to drastically reduce its free expansion energy. Indeed, metrological applications suffer predominantly from expansion size effects of the interrogated samples that could limit the excitation efficiencies in atom-light interactions or cause leading systematic effects such as wavefront aberrations[25,46]. Attempts to reach sub-nK energies were reported using magnetic levitation techniques[47] or adiabatic decompression[5,48]. To further reduce the quantum gas energy, we turn here to atomic lensing using the delta-kick collimation (DKC) techniques proposed in refs. 49,50 and recently implemented in refs. 41,51,52. The lensing sequence consists of releasing the quantum gas from the trap for a certain free evolution time and shortly re-flashing the trapping potential for a specific lens duration to slow down the expansion of the atomic cloud. Here we use trap A $(\omega_x, \omega_y, \omega_z) = 2\pi \cdot (25.2, 109, 110)$ Hz and apply it to lens the released BEC matter wave after rescaling its trapping frequencies by a factor 1/4. This rescaling is mandatory to prevent impractically short lensing durations. The quasi-cylindrical symmetry of trap A enables the simultaneous collimation along the $y$- and $z$-axis with a single lensing pulse yielding very low overall 3D expansion energies.

In order to avoid momentum kicks for the center-of-mass and shape distortions, the atom cloud must be located at the minimum of the lensing potential during the delta-kick. We therefore verify the stability of our BEC source for the optimal initial hold time $t_{hold,i} = 2.4$ ms determined in Fig. 2 and record the release velocity out of trap A while scanning the final hold time $t_{hold,f}$. The data (see Supplementary Fig. 5) shows an excellent consistency with our theoretical model despite being collected over several months in a mobile laboratory experiencing sizable orientation and altitude changes. For a final hold time of 24 ms, the release velocity is minimal and the BEC has moved only by 0.2 ± 5.9 μm after 20 ms of free fall. In addition, this timing corresponds to a low expansion energy (200 ± 27 pK) in the $x$-direction triggered by the collective excitations of the quantum gas similar to the findings of refs. 38,41. This is important in order to constrain the atomic density to a detectable level for several 100 ms after release, even though no appreciable collimation is expected in the $x$-direction due to the weak trap frequency.

Figure 4 shows the results of our atomic lensing experiments in the $z$-direction for a lens duration of 1.8 ms (green) contrasted to the free evolution after release (red). Indeed, in this direction we have reduced the expansion energy by roughly a factor 70 from 3.6 ± 0.2 nK down to only 52 ± 10 pK. Thanks to the symmetry of the trap frequencies, we infer a similar performance in the $y$-direction. Thus, together with the slow expansion along $x$, the total 3D kinetic energy of the source reaches about 100 pK compatible with the landmark several-second, space operation. This performance compares well with ground-based 2D delta-kick experiments[52] and could be enhanced to match the achievement of state-of-the-art 3D delta-kick collimation[41] at the cost of adding an extra preparation step further reducing the velocity in the weakly trapped direction through collective excitations engineering. The longest free evolution time was limited to roughly 350 ms due to technical constraints of the apparatus (see Methods section). The experimental results were juxtaposed to an ab-initio model (see Methods section) that describes the quantum mean-field dynamics of the condensed gas in all three dimensions throughout the

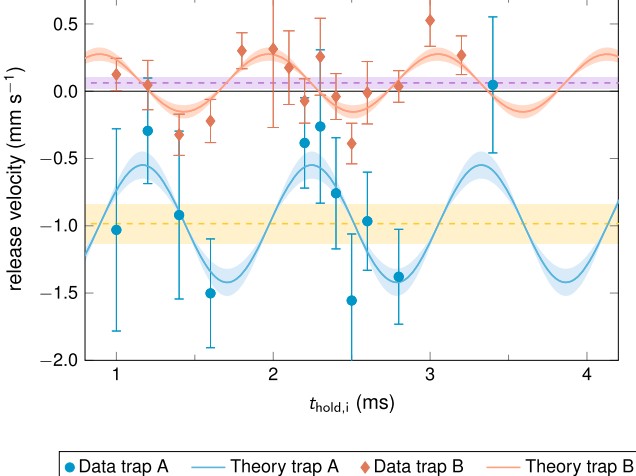

**Fig. 3 | Tunable and robust trap release of a BEC.** Release velocity of a transported BEC as a function of the initial hold time $t_{hold,i}$ for the two transport ramps A (blue) and B (red) described in the caption of Fig. 2. Both transports allow for tunable release velocities between −1.5 mm s$^{-1}$ and +0.5 mm s$^{-1}$ with a minimal uncertainty of 0.117 mm s$^{-1}$ for $t_{hold,i} = 2.8$ ms (trap B) representing experimental noise. The release velocities are subject to a trap-dependent systematic shift (yellow and purple dashed lines) given by $\Delta v_A = -0.98 \pm 0.15$ mm s$^{-1}$ and $\Delta v_B = +0.062 \pm 0.045$ mm s$^{-1}$ caused by the finite switch-off time of the trap. The data (blue circles and red diamonds) is consistent with an ab-initio theoretical model (blue and red lines) based on a chip simulation and mean-field BEC dynamics (see Methods). The error bars and shaded areas show the 1σ-confidence bounds of the fits and the model, respectively.

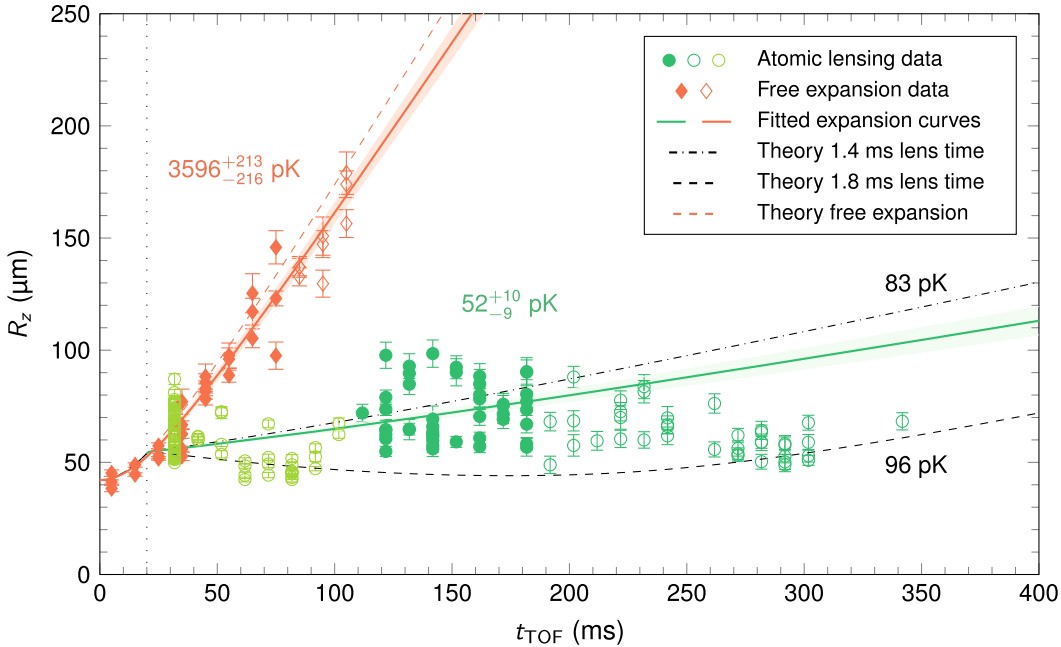

**Fig. 4 | Delta-kick collimation of a BEC to the 50 pK energy level.** Thomas-Fermi radius in the $z$-direction $R_z$ of a free expanding BEC (red diamonds) released from trap A compared to the case where the final trap was switched on again for 1.8 ms (green dots) 20 ms after release (dotted vertical line) with rescaled angular frequencies (by a factor 1/4) to lens the BEC. The atomic lens removes kinetic energy from the system yielding an expansion energy of only $52 \pm 10$ pK compared to $3.6 \pm 0.2$ nK for the non-lensed ensemble. The expansion energies are determined by fitting theory curves (solid lines) to the data (filled dots). Shaded areas show the full sequence including the transport, the release, the delta-kick collimation at $t_{TOF} = 20$ ms and the final free expansion. As a result, for different lens durations (see Supplementary Fig. 6), we have noted a delay in the magnetic field switch-off, globally quantified to a maximum of 0.4 ms. The region between the dashed and dash-dotted black lines reflects this uncertainty of the experimental arrangement. Within this confidence area, we observe an excellent agreement with our theoretical predictions for the variety of lens regimes that we applied.

$1\sigma$-confidence bounds of the fits and error bars reflect the single-shot detection noise. Comparison with a realistic 3D simulation of the Gross–Pitaevskii equation (dashed line) reveals that the effective lens is weaker than expected due to finite switching times. Scanning the lens duration (see Supplementary Fig. 5) leads to an upper bound of the expansion dynamics by a 0.4 ms shorter lens (dash-dotted line) with the corresponding expansion energies given in black. Data shown by empty symbols are excluded from the fits due to split clouds at short times and low densities at long times (see Methods section).

## Discussion

The experiments conducted here in a modular, multi-user quantum gas laboratory represent an important toolbox for cold atoms in space. In contrast to ground-based research, several non-trivial hurdles had to be tackled including the inability to proceed to hardware changes, the remote operations and executions of a limited number of sequences interfacing with distant JPL scientists and engineers and the delayed retrieval of data. Going beyond the previous demonstrations, the challenge here was to actually take advantage of the lack of gravity to quantitatively surpass Earth-bound performances such as the demonstrated shuttling and positioning precision of ensembles of neutral atoms. More importantly, and driven by the promises of space-based cold atom technology in several fields such as secure quantum communication, quantum computation or quantum sensing, it was necessary to demonstrate that it is possible to realize an atomic source with a great accuracy in terms of its quantum state engineering. Enabled by our results, space experiments are expected to boost the sensitivity of quantum sensors through the unrivaled free drift time they will offer. This exquisite sensitivity puts, however, quite demanding requirements on the preparation of initial states. To give a specific example, a dual-species interferometer testing the universality of free fall at the state of the art[53] would require the positioning uncertainty to be not larger than 1 μm and the expansion energy to be at the order of few tens of pK. These two stringent requirements were

fulfilled by the quantum state engineering that was achieved here which in turn demonstrates the feasibility of ambitious fundamental physics missions in space such as the satellite test of the Einstein equivalence principle STE-QUEST at the $10^{-17}$ level[29] competing for a medium-class mission opportunity within the ESA's science programme[54].

This chip-integrated atomic source therefore already satisfies the most stringent requirements set for future space missions based on precision atom interferometry[28,29,55,56]. We also anticipate an extensive utilization of such an atomic source preparation in various other application fields such as quantum information processing[18], satellite gravimetry[28], matter-wave optics[15], fundamental physics tests[42], gravitational wave detection and space quantum communication[33,34].

Moreover, we expect a great improvement of results with the next generation of CAL and the envisioned BECCAL[57] payloads. Indeed, further increasing the atom numbers ($10^5$ instead of $10^4$) in the BEC would dramatically improve the signal strength, while more flexible current controllers would enable even smoother transport ramps (ramps of a few hundred discrete steps instead of 80 in this work). Finally, dedicated studies with atoms in the magnetically insensitive Zeeman states will allow for undisturbed freely floating atomic clouds. These purely technical improvements could already make it possible to access positioning accuracies at the nm level and observation times of several seconds compatible with the most ambitious quantum technologies applications in space[58].

In summary, we have performed state-of-the-art quantum-state engineering of a quantum degenerate gas of neutral atoms in the Earth-orbiting research laboratory CAL. The protocols applied were robust enough to allow the successful, rapid transport of a quantum gas with near 70 nm positioning accuracy, its tunable controlled release with velocities known at the 100 μm s⁻¹ level and reduced its expansion energy to about 50 pK. The high fidelity in reaching this state optimization and its stability over months and millions of km of

operation on a 90-minute orbit inaugurate the metrological exploitation of space-embarked quantum technologies based on neutral atoms.

## Methods

### Cold Atom Lab facility

CAL is a NASA multi-user BEC facility that was launched to the ISS in May 2018. It was built and operated by a team of scientists at the Jet Propulsion Laboratory in Pasadena, California, USA. As this team detailed in refs. 5,59, the facility allows the study of $^{87}$Rb BECs in microgravity in an atom-chip-based device. The experiments reported in this article are efforts by the Consortium for Ultra-Cold Atoms in Space (CUAS). The final evaporation of the atomic cloud takes place in a tightly confining trap generated by the combination of magnetic fields resulting from five different currents applied to two chip structures as well as three pairs of Helmholtz bias coils. In order to increase the robustness of the quantum state engineering proposed by the CUAS team, the BEC was loaded into a simplified trap generated by solely two currents, one sent through a Z-shaped chip wire, the other, $I_{coil}$, injected in an external pair of Helmholtz bias coils oriented along the $y$-axis (parallel to the chip surface). The shuttling of the atoms, in this simplified trap, is controlled solely by the value of the current $I_{coil}$. These traps were calibrated in a dedicated campaign as explained in the next point.

Although initially featuring more than 10,000 BEC atoms[5], degradation of the alkali metal dispenser performance limited the number of condensed atoms to a maximum of 4000 when most of the CUAS experiments reported here were performed. Consequently, free evolution times were limited to 400 ms even for appropriately lensed ensembles due to decreasing particle densities. Moreover, the residual magnetic field gradients present even when all coils and chip structures were switched off[5,48], led to relevant accelerations of the atoms in the $m_F = 2$ hyperfine sublevel so that the BEC ultimately moved out of view of the imaging system. However, our analysis showed that any corresponding magnetic field curvatures were negligible for the dynamics of the atom clouds. Attempts to transfer the atoms to the magnetic insensitive $m_F = 0$ sub-level were prevented by low atom numbers and the limited total operation time of the CAL device. Later on, the core part of the apparatus, including the atom chip, was exchanged leading to a new magnetic field geometry and thus preventing the further continuation of the experiments reported in this article.

### Calibration efforts for the chip model

In order to design efficient non-adiabatic transport ramps, a precise calibration of the applied chip and coil currents and the resulting trap positions and frequencies is mandatory. By setting up a 3D model of the atom chip wires as well as the Helmholtz coils and solving the Biot-Savart law for these current-carrying structures, we were able to compute the value of the magnetic field at the position of the atoms. The predictions of this chip model were compared with data of the in-trap oscillation of the atomic cloud transported to different positions (see Supplementary Fig. 1). Here the initial BEC was diabatically transported to an intermediate trap by linearly ramping down the current in the $y$-coils to induce center-of-mass sloshing which was then recorded by varying the hold time in that trap. By fine-tuning the parameters of the chip model, we were then able to calibrate our theoretical model with the measured data. The ramp sequences discussed in the main text are based on this chip trap modeling.

### Fitting routine

All data shown in this article is based on resonant absorption imaging along the $y$-axis and fitting of the obtained 2D-density distribution ($x$-$z$-plane) either by a Thomas-Fermi profile (free expansion times beyond 80 ms) or by a bimodal fitting routine (calibration, STA

characterization and short free expansion times) consisting of a Gaussian background for the thermal part and a Thomas-Fermi profile for the condensed fraction. Here we focus solely on the results of the condensed part of the atom cloud since the thermal part typically expands much faster than the BEC and already gets too dilute to be measurable after a time-of-flight of several tens of ms. Consequently, we measure the spatial extent of the BEC in terms of the Thomas-Fermi radius. Our science experiments were typically performed with 2000–4000 condensed atoms and a BEC fraction of 10–25%. During the calibration measurements the BEC fraction was below 10% and the number of condensed atoms reached up to 12,000.

### Shortcut-to-adiabaticity protocol and robustness study

The STA protocol developed for harmonic traps in ref. 38 and implemented in CAL's atom chip, assumes a BEC initially at rest with vanishing center-of-mass velocity and acceleration located at the minimum of the initial trap. The CAL-delivered quantum gas is however oscillating with an amplitude of $0.22 \pm 0.05 \, \mu m$ in the $z$-direction (926 Hz frequency) in the starting trap and therefore never matches these theoretical conditions. Nevertheless, the STA ramp employed here is robust against an initial position offset (see Supplementary Fig. 2) and the starting point with vanishing velocity is found by scanning the ramp starting time. Moreover, the atom chip traps are harmonic only close to the minimum such that during the diabatic transport, the quantum gas explores anharmonic regions of the potential. On the other hand, low amplitude oscillation ramps require fast current changes which could not be implemented experimentally due to the limited amount of discretization steps. This limit leads to an important mismatch between the theory ramps and their experimental implementations and therefore to high residual amplitude oscillations in the final trap. A trade-off was found and led to the proposal of ramps of 100 and 150 ms duration - and not shorter - in order to limit these two effects. A robustness study was carried out in order to evaluate the residual oscillations in the final trap as a function of the initial position of the transported atomic cloud. The result of this analysis is shown in Supplementary Fig. 2 which generalizes the theory of ref. 38 to transport traps with additional cubic terms (the most important anharmonic potential components for atom chips) and to account for the effects of the discretization of the current $I_{coil}$ in 80 steps as communicated to the operations team. This discretization in a finite number of steps is a clear limit for the implemented ramps. If it would technically be possible to code a ramp with 200 steps, one would reach the nm positioning uncertainty predicted by the robustness study. By tuning the initial hold time, one could obtain the optimal results (better than 1 μm residual oscillation amplitude) at the turning point of the initial oscillation ($z_i = 0.2 \, \mu m$, $v_i = 0$) as predicted by the adapted theory described here. It is interesting to note that the STA ramps found here do not have a complex shape and could be approximated by a sigmoid with an adjusted inflection point and slope. Finding such a sigmoid would however require a large number of experiments exploring a wide range of these two parameters. Conversely, the STA ramp is found immediately, its robustness is evaluated theoretically and is therefore a powerful quantum engineering tool in an environment where the number of experimental shots is limited.

### Theory of the transport and release dynamics

The theoretical model of the in-trap amplitude and release velocity of the BEC is based on the solution of Newton's equation of motion for the displacement of a particle in a harmonic potential, with a time-dependent trap frequency $\omega_z(t)$ and position $z_{trap}(t)$ known at every position thanks to a complete atom chip simulation calibrated to the experiment (see the calibration section above).

We measured through sloshing experiments in the initial trap (Fig. 1c), that the condensate is oscillating with an amplitude of $A_i = 0.22 \pm 0.05 \, \mu m$. We thus consider an oscillation in the initial trap

with this amplitude as the initial conditions

$$z_{\mathrm{at}}(0; t_{\mathrm{hold,i}}) = A_i \cos[\omega_{z,i} t_{\mathrm{hold,i}} + \varphi_i],  \quad (1)$$

and

$$z'_{\mathrm{at}}(0; t_{\mathrm{hold,i}}) = -A_i \omega_{z,i} \sin[\omega_{z,i} t_{\mathrm{hold,i}} + \varphi_i],  \quad (2)$$

for the equation of motion

$$z''_{\mathrm{at}}(t; t_{\mathrm{hold,i}}) + \omega_z^2(t)\left[z_{\mathrm{at}}(t; t_{\mathrm{hold,i}}) - z_{\mathrm{trap}}(t)\right] = 0,  \quad (3)$$

determining the position $z_{\mathrm{at}}(t; t_{\mathrm{hold,i}})$ of the atomic cloud. By solving this equation of motion, we obtain the final position $z_{\mathrm{at}}(t_f; t_{\mathrm{hold,i}})$ and the release velocity $z'_{\mathrm{at}}(t_f; t_{\mathrm{hold,i}})$ of the atoms, which are periodic functions with respect to the initial hold time $t_{\mathrm{hold,i}}$ with the same frequency as the one of the initial trap.

The phase of the initial oscillation $\varphi_i$ is adjusted so that the release velocity variation is in phase with the data. The theoretical offset of the oscillation of the condensate velocity is different from the experimental offset. This shift is given by a sinusoidal fit of the experimental data and is attributed to a kick induced by the chip's currents switch-off.

The oscillation amplitude $A_f(t_{\mathrm{hold,i}})$ in the final trap of frequency $\omega_z(t_f)$ is deduced by the motion of a particle in a harmonic potential. It is obtained from the position and the velocity of the center of mass of the condensate at the end of the transport by the following formula

$$A_f(t_{\mathrm{hold,i}}) = \sqrt{[z'_{\mathrm{at}}(t_f; t_{\mathrm{hold,i}})/\omega_z(t_f)]^2 + [z_{\mathrm{at}}(t_f; t_{\mathrm{hold,i}}) - z_{\mathrm{trap}}(t_f)]^2}. \quad (4)$$

The shaded areas of the theoretical plots are the result of calculations taking the maximum and minimum values of the oscillation amplitude in the initial trap, which are given by the experimental confidence bounds on the amplitude.

### Delta-kick collimation analysis

The atomic lens modeling relies on a mean-field approach solving the Gross–Pitaevskii equation in 3D for a time-dependent trap sequence starting with the initial 2-current trap and going through the whole transport protocol.

In order to determine the expansion velocities of the freely expanding and lensed BECs, we fit theory curves (solid lines in Fig. 4 and Supplementary Fig. 6) to the data sets (filled circles). These expansion curves are based on effective scaling solutions[60–62] for harmonically trapped BECs and take into account the mean-field dynamics giving rise to a non-trivial short-time behavior and a linear expansion for long free evolution times. These scaling solutions agree very well with full numerical simulations of the Gross–Pitaevskii equation systematically lying between the two bounds represented in Fig. 4 and Supplementary Fig. 6 by the black dashed and dash-dotted lines. In case of lensed atoms, we varied the duration of the lens to obtain the optimal fitting expansion curve while in case of free expanding atoms and for data along the $x$-direction the frequency of the release trap in the relevant direction was varied. The slope $dR/dt$ of the resulting curves in the long-time limit is used to calculate the expansion energy of the cloud in terms of an expansion energy $k_B T/2 = m(dR/dt/\sqrt{7})^2/2$, where the factor $\sqrt{7}$ relates the Thomas-Fermi radius to the standard deviation of the cloud, $m$ being the mass of a $^{87}$Rb atom and $k_B$ the Boltzman constant. The error bars are given by the 1$\sigma$-confidence bounds of the fits.

During the analysis of the atomic lensing experiments, the following effects were identified and taken into account: (i) Shortly after the atomic lens is applied, the BEC was split into two individual clouds ($m_F = 1$ and $m_F = 2$ hyperfine sublevels) by non-adiabatic spin-flip transitions due to magnetic field switching similar to what was

observed in ref. 48. In our case, the splitting appeared in the $x$-direction and the majority of the atoms remained in the $m_F = 2$ state. Moreover, only the atomic lens experiments were affected but not the pure free expansion. After a total time-of-flight of 100 ms, the two clouds were well separated allowing to individually fit the $m_F = 2$ cloud for the rest of the dynamics. As a consequence, the data points for times $t_{\mathrm{TOF}} \le 100$ ms (light green open circles in Fig. 4) were fitted to overlapping clouds rendering a precise determination of the cloud size challenging, especially in the $x$- direction. (ii) For long expansion times our modeling predicted densities below the experimentally detectable minimum of $2 \cdot 10^{11}$ atoms/m² so that the data points beyond this density threshold (dark green open circles) were not considered for fitting the expansion curves. This cutoff depends on the lens duration and increases from 120 ms to 190 ms for Supplementary Fig. 6a–d. (iii) The determination of the size of the atom cloud was limited to a minimum of roughly 40 μm in the $z$-direction although the initial size of the BEC was less than 10 μm. Thus, we included a correction factor $R_{\mathrm{reso}}$ for the theoretically determined sizes $R_z$ by considering the expression $(R_z^2 + R_{\mathrm{reso}}^2)^{1/2}$ for the theory curves. Comparison with the short time behavior of the free expansion dynamics resulted in the value $R_{\mathrm{reso}} = 42$ μm used for all data sets.

## Data availability
The datasets generated for and analyzed in this paper are available from the corresponding author upon reasonable request. All NASA CAL data are on a schedule for public availability through the NASA Physical Science Informatics (PSI) website (https://www.nasa.gov/PSI).

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

## Acknowledgements

Designed, managed and operated by Jet Propulsion Laboratory, Cold Atom Lab is sponsored by the Biological and Physical Sciences Division of NASA's Science Mission Directorate at the agency's headquarters in Washington and the International Space Station Program at NASA's Johnson Space Center in Houston. This work is supported by NASA through the Jet Propulsion Laboratory Research Service Agreements including RSA 1616833a and the DLR Space Administration with funds provided by the Federal Ministry for Economic Affairs and Climate Action (BMWK) under grant numbers DLR 50WM1861-2 (CAL) (M.M., P.B., W.P.S., N.G., E.M.R., A.P.), 50WM2245-A/B (CAL-II) (P.B., W.P.S., N.G., E.M.R., A.P.), 50WP1705 (BECCAL) (M.M., W.P.S.), 50WM2060 (CAR-IOQA) (N.G., E.M.R.) and 50WM2263A (CARIOQA-GE) (N.G., E.M.R.) and is funded by the Deutsche Forschungsgemeinschaft (DFG, German Research Foundation) under Germany's Excellence Strategy—EXC-2123 QuantumFrontiers—390837967 (E.M.R.) and through the CRC 1227 (DQ-mat) within Project Nos. A05 (N.G.), B07 (E.M.R.). N.G. acknowledges funding from "Niedersächsisches Vorab" through the Quantum- and

Nano-Metrology (QUANOMET) initiative within the project QT3 and H.A. through "Förderung von Wissenschaft und Technik in Forschung und Lehre" for the initial funding of research in the new DLR-SI Institute. A.P. and E.C. thank the Mésocentre computing center of CentraleSupélec and École Normale Supérieure Paris-Saclay supported by CNRS and Région Île-de-France (http://mesocentre.centralesupelec.fr/) for HPC resources.

## Author contributions

N.G., M.M., R.C., and W.H. gauged the facility, proposed the experiments and coordinated with JPL scientists. M.M., A.P., P.B., and H.A. evaluated the data. N.G., R.C., A.P., and E.C. carried out the simulations. J.R.W. and R.J.T. communicated the consortium's sequences to the ISS and executed the science campaigns. N.G., M.M., W.P.S., E.M.R., and N.P.B. wrote the manuscript with contributions from all authors. N.P.B. is the principal investigator and Director of the CUAS consortium. All authors read, edited and approved the final manuscript.

## Funding

## Competing interests

The authors declare no competing interests.
