## [Peer Review File · Nature Communications]

A space-based quantum gas laboratory at picokelvin energy scalesEditorial Note: This manuscript has been previously reviewed at another journal that is not operating a transparent peer review scheme. This document only contains reviewer comments and rebuttal letters for versions considered at *Nature Communications*.

REVIEWERS' COMMENTS

Reviewer #1 (Remarks to the Author):

The authors have answered well all my concerns and the revised manuscript is clearly suitable for publication in Nature Comm.

Reviewer #2 (Remarks to the Author):

This work reports on precision state preparation on the cold atom research laboratory (CAL) onboard the ISS.

The paper is well written and interesting and on the forefront of current research. The authors have demonstrated fast transport over a large distance, precision positioning and reduction of the temperature to below 100pK. All of these are important and impressive steps for atom interferometry in space. The demonstrated data is of high quality, well presented and matches well the applied theory.

The authors claim that the system will not be used in the current platform. To what extent the developed techniques are therefore directly useful or transferable to earth-bound interferometers – is unclear to me and might be difficult to judge.

I find this work exciting and readers will be interested to hear about progress of cold atom research on board the ISS.

The authors have demonstrated in high quality work improved (in the sense of going beyond the state of the art) atomic transport, improved cooling and precise localisation of the atomic cloud – these methods are not completely new but relevant for future precision measurements based on atom interferometry in space.

Response To Referee Comments

Final Resubmission of Nature Communications manuscript NCOMMS-22-36464-T

"A space-based quantum gas laboratory at picokelvin energy scales,"

Dear Editors of Nature Communications, Dear Referees,

We thank the referees for critically and enthusiastically reviewing our manuscript. There was only one open comment from one of the Referees:

“The authors claim that the system will not be used in the current platform. To what extent the developed techniques are therefore directly useful or transferable to earth-bound interferometers – is unclear to me and might be difficult to judge.”

The question of whether our results, accomplished on-orbit, are directly useful to Earth-bound interferometers is addressed in our resubmission. Specifically, on page 3, in the third full paragraph beginning “These protocols...,” we explain that the near absence of gravity on orbit decouples the transport protocol degrees of freedom allowing us to achieve simple efficient transport that would not be possible on Earth. In the next paragraph we further point out that one of the traps used in the atom shuttling study is only usable on-orbit where the residual gravitational pull of the Earth is so weak. While, as noted, these results will not be used directly using the current CAL device aboard the International Space Station, our accomplishments will be applied to experiments planned using the next upgrade of CAL and on future BEC space platforms.

Thank you very much,

Nicholas P. Bigelow, Naceur Gaaloul and Matthias Meister, on behalf of the Authors